# Antimicrobial Efficiency of *Pistacia lentiscus* L. Derivates against Oral Biofilm-Associated Diseases—A Narrative Review

**DOI:** 10.3390/microorganisms11061378

**Published:** 2023-05-24

**Authors:** Egle Patrizia Milia, Luigi Sardellitti, Sigrun Eick

**Affiliations:** 1Department of Medicine, Surgery and Pharmacy, University of Sassari, Viale San Pietro 43, 07100 Sassari, Italy; 2Dental Unit, Azienda Ospedaliero-Universitaria di Sassari, 07100 Sassari, Italy; 3Department of Periodontology, School of Dental Medicine, University of Bern, Freiburgstrasse 3, 3010 Bern, Switzerland; sigrun.eick@unibe.ch

**Keywords:** plant-derived products, polyphenol extracts, medicinal plants, periodontal disease, caries, *Candida*

## Abstract

*Pistacia lentiscus* L. (PlL) has been used for centuries in traditional medicine. The richness in antimicrobial biomolecules of Pll derivates can represent an alternative to chemically formulated agents used against oral infections. This review summarizes the knowledge on the antimicrobial activity of PlL essential oil (EO), extracts, and mastic resin against microorganisms being of relevance in oral biofilm-associated diseases. Results demonstrated that the potential of PlL polyphenol extracts has led to increasing scientific interest. In fact, the extracts are a significantly more effective agent than the other PlL derivates. The positive findings regarding the inhibition of periodontal pathogens and *C. albicans*, together with the antioxidant activity and the reduction of the inflammatory responses, suggest the use of the extracts in the prevention and/or reversal of intraoral dysbiosis. Toothpaste, mouthwashes, and local delivery devices could be effective in the clinical management of these oral diseases.

## 1. Introduction

The most prevalent oral diseases are strongly associated with a loss of microbial homeostasis and biofilm development.

Due to a microbiota involving carbohydrate-fermenting *Streptococcus mutans* and species of the genera *Actinomyces*, *Lactobacillus*, *Dialister*, *Eubacterium*, *Olsenella*, *Bifidobacterium*, *Atopobium*, *Propionibacterium*, *Scardovir*, *Abiotrophia*, *Selenomonas*, *Veillonella*, and *Candida albicans* biofilms on the tooth surfaces may cause dental caries [1]. At the gingival and subgingival margins, biofilm accumulation may lead to gingivitis, which occurs as an inflammatory infiltrate as a consequence of an imbalance of the host’s immune response. A loss of eubiosis may increase the percent of proteolytic and anaerobic bacteria, *e.g.*, *Porphyromonas gingivalis*, *Tannerella forsythia*, *Treponema denticola*, *Prevotella intermedia*, all characterizing the periodontal disease [2,3,4]. Also, peri-implant diseases may develop due to the accumulation of bacteria [5]. Further, *Porphyromonas*, *Fusobacterium*, and *Prevotella* ssp. together with *Tannerella forsythia* and *Veillonella* spp. represent the most related oral bacteria in intraoral halitosis with the production of volatile sulfur compounds (VSCs) [6]. The compounds derived by the activity of Gram-negative anaerobic bacteria are supported by Gram-positive bacteria and fungi [7,8]. In comparison, Gram-negative anaerobic rods and proteolytic bacteria biofilm recur in endodontic and periapical infections related to pulp necrosis [9]. Regarding the yeast, *Candida* biofilm, with a high prevalence of *C. albicans* and/or *C. glabrata* in immunocompromised patients, leads to oral candidiasis that may have severe consequences under these pathological conditions [10]. Finally, Primary Herpetic Gingivostomatitis (PHG) and herpes labialis represent the most common clinical evidence of oral viral infections by the Herpes Simplex Virus type-1 (HSV-1) [11].

Accordingly, antiseptics are widely used to prevent and treat the above-mentioned conditions. Chlorhexidine digluconate is still the gold standard among antiseptics [12]. Nevertheless, tooth staining, calculus formation, and high cytotoxicity against human fibroblasts and osteoblasts have been described as side effects when using chlorhexidine [13,14]. Antibiotics, *e.g.*, amoxicillin, metronidazole, and azithromycin, are applied in severe cases of periodontal disease, reporting side effects like nausea, vomiting, diarrhea, and others [15]. However, resistance to antimicrobials, which is associated with the use of these substances, has become a global problem.

Plant-derived substances have been considered to overcome the issues of antiseptics and antibiotics, plant-derived substances are highly discussed [16]. PlL, belonging to the *Anacardiaceae* family [17], has been used for centuries in traditional medicine. Almost the whole plant, including the leaves, fruits, wood, and mastic resin, has been utilized as a remedy for a large variety of diseases (Figure 1). The richness in antimicrobial and anti-inflammatory biomolecules might represent an alternative to chemically formulated therapies.

Harsh growing conditions, dryness, and a warm environment influence on the genotype and richness of secondary metabolites of PlL. They are mainly represented by terpenoids in the EO, and flavonoids, phenolic acids, and derivatives in the polyphenols mixture of the extracts [17,18].

In ethno-pharmacology, different popular formulations of PlL, including the EO, the boiled extracts, the poultices of blossoms and leaves, patches of woods, and mastic resin, have been used [19,20,21,22,23,24,25]. Furthermore, PlL has been administered in the form of smoke, obtained by burning or boiling the soft wood and leaves, particularly in the cases of osteoarthritis, bronchitis, and allergies [21,25]. Moreover, using leaf extracts for beverages and mouthwash or direct chewing of the soft stems, leaves, or mastic is still used to antagonize toothache and gingival inflammation [24,25,26,27]. Additionally, powdered mastic can be applied to allow wound healing [19,20] and to antagonize gram-positive bacteria, including *Staphylococcus aureus* methicillin-resistant species [28], while the leaves EO acts with high antibacterial capacity against gram-negative rods as *Escherichia coli*, and *Pseudomonas aeruginosa* [29].

Regarding inflammation, the beneficial effect of PlL may be related to the anti-oxidant capacity [30,31,32,33] and the antagonism to inflammatory cytokines, i.e., interleukin (IL)-1β, IL-6, and tumor necrosis factor-α (TNF-α) [32,34]. Additionally, proven activity toward the arachidonic acid cascade, particularly against COXs and 5-LOX enzymes has also been demonstrated [35,36].

Anti-inflammatory and anti-ROS potency, and the capacity to target oral bacteria have led to the proposal of PlL derivates as a natural antimicrobial agent against oral diseases [18,36,37,38].

Given the above considerations, after describing the major chemical constituents of PlL, the following overview discusses in-vitro studies on the antimicrobial activity of the EO, extracts, and mastic against microorganisms being of relevance in oral biofilm-associated diseases. Additionally, the clinical studies conducted to ascertain the scientifical value of PlL were summarized. Finally, we discussed the potential of PlL secondary metabolites as an oral agent and the greater indications they may have in preventing and treating oral diseases.

## 2. Phytochemical Constituents of *Pistacia lentiscus*

The growth environment, seasonability of harvesting, and type of material (edible or not edible parts of the plant) influence chemical differences of PlL oils and extracts between the Mediterranean regions [17] (Table 1).

PlL EO is constituted by a mixture of terpenes and terpenoids, mainly monoterpenes and sesquiterpenes. They are responsible for the characteristic smell and flavoring of the plant and derivates [39]. Among them, *α*-pinene, terpinene, caryophyllene, limonene, and myrcene are the most recurrent compounds in PlL EO of leaves [17,40,41,42,43]. Additionally, up to 64 fractions have been identified in the leaves EO fingerprint, among which some terpenoids are constituent fractions of *Cannabis sativa* [44] and are reported as “non-cannabinoid terpenoids.” They can induce cannabinoid-like properties in immune-mediated inflammatory and autoimmunity diseases, neuroinflammatory, neurological, and neurodegenerative conditions, and exert anti-infective and anti-cancer properties [45,46]. Furthermore, given the prevalent fractions of monoterpenes and oxygenated sesquiterpenes, different EO chemotypes merge in the regions [17,36]. In North Sardinia, Italy, the recurrent higher amount of *α*-pinene (16.9–19.5%) and terpinen-4-ol (7.7–16.5%) in comparison to the other compounds, allowed the classification of the leaves EO as *α*-pinene/terpinen-4-ol chemotype [36]. While the Corsican chemotype of the oil is expressed by three main phenotypes: the first is *α*-pinene/terpinen-4-ol; the second is terpinen-4-ol/limonene; and the third is myrcene-rich (88%) [17].

PlL EO from Egypt is rich in δ-3-carene (65%) [47], while the monoterpen terpinen-4-ol, together with *α*-pinene and the sesquiterpene myrcene are among the higher represented fractions in the chemistry of the EO from Spain, Morocco, and Turkey [40,41,43].

Regarding the oils of mastic resin, *α*-pinene (65–86%) and *β*-myrcene (3%) are the major fractions [27,48,49,50]. While limonene, sabinene, and myrcene, in addition to 𝛼-pinene, are the main representative compounds of the fruits EO [51]. the richness in anthocyanins is highly characteristic in the berries oil, conferring to the agent’s antioxidant ability [52].

Comprehensive studies have been conducted on the methanol and alcohol extracts of PlL leaves, alongside investigating their chemical characteristics. The high concentration of polyphenols in the extract is extremely attractive from a beneficial point of view. Polyphenols can antagonize oxidative stress and inflammation, exerting anti-cancer, anti-obesity, and anti-diabetic character, further increased by cardiovascular, hepatoprotective and osteoporotic preventive effects [53,54]. Additionally, polyphenols have high value as anti-infective compounds [32,55,56,57].

In the methanol extracts of PlL leaves from plants growing in Algeria, a number of 46 different molecules were identified [58]. Flavonoids, phenolic acids, and their derivatives were the most abundant, including myricetin glycoside, catechin, *β*-glucogallin, and quercitrin gallate. High doses of polyphenols in the ethyl acetate and methanol extracts of leaves were found by Romani and coworkers [59]. In that research, PlL polyphenols reached up to 7.5% of leaf dry. Furthermore, three major classes of secondary metabolites were recognized in the extract (i) gallic acid and galloyl derivatives of both glucose and quinic acid; (ii) flavonol glycosides, i.e., myricetin and quercetin glycosides; and (iii) anthocyanins, namely delphinidin 3-O-glucoside and cyanidin 3-O-glucoside. The leaf extract had greater content of phenols and flavonoids than that of the fruit, which conversely exhibited a higher number of tannins. A correlation between the polyphenols content and biological capacity of PlL extracts has been further strengthened by Garofulić and coworkers using a sophisticated microwave-assisted extraction (MAE) [60]. By MAE, the authors were able to isolate an increasing proportion of biomolecules, including the more heat-sensitive complex structures of flavonoid glycosides, procyanidins, and tannins, which proportionally raised the antioxidant capacity of the material. Very recently, a phytochemical study further confirmed the link between the content of phenol, flavonoid, and tannin to the antioxidant activity of extract [61].

Concerning the antimicrobial mechanism, it is proposed that the activity of polyphenols involves a membrane-disrupting mechanism, together with the inhibition of cell envelope synthetases, the nucleic acids synthetases, the bacterial virulence, the efflux pump, NADH-cytochrome C reductase, and ATP synthetase. All these cell disorders, further increased by an impairment of Quorum Sensing, lead microorganisms to inability to form biofilms [62,63,64].

**Table 1 microorganisms-11-01378-t001:** Chemical profiles of *Pistacia lentiscus* L. Adapted from Milia et al., 2021 [18].

**Essential Oil**				
**Plant Material**	**Origin**	**Main Components**	**Test Assays**	**Ref.**
Leaves	Sardinia (Italy)	*α*-pinene (14.8–22.6%), terpinen-4-ol (14.2–28.3%), *β*-myrcene (1.0–18.3%) and other minor fractions.	GC-MS	[17]
Mastic gum	Greece	*α*-pinene (65–86%) and *β*-myrcene (3%)	GC-MS	[27]
Leaves	Eastern Morocco	Taforalt and Saidia areas: limonene, *α*-pinene, *α*-terpineol, and *β*-caryophyllene. Laayoune and Jerada areas: myrcene, and *β*-caryophyllene.	GC-MS	[29]
Dried leaves		δ-cadinene (17.04%), *α*-amorphene (10.32%), δ-germacrene (9.01%) and other minor fractions.	GC-MS	
Leaves	Sardinia (Italy)	*α*-pinene, *α*-thujene, camphene, sabinene, *β*-pinene, myrcene, *α*-phellandrene, *α*-terpinene, para-cymene, *β*-phellandrene, trans-ocemene, γ-terpene, terpinolene, 2-nonanone, linalool, isopentyl isovalerate, terpin-4-ol, *α*-terpiniol, and others.	GC-MS	[34]
Leaves	Sardinia (Italy)	Germacrene D (19.9%), *β*-caryophyllene (6.6%), *α*-pinene (6.3%) and other minor fractions.	GC-MS	[35]
Leaves	Sardinia (Italy)	*α*-pinene (16.9%), terpinen-4-ol (16.5%), sabinene (7.8%) and other minor fractions.	GC-MS	[36]
Leaves	Turkey	Terpinen-4-ol (29.9%), *α*-terpineol, (11.6%), limonene (10.6%) and other minor fractions.	GC-MS	[40]
Leaves	Morocco	Myrcene (39.2%), limonene (10.3%), *β*-gurjunene (7.8) and other minor fractions.	GC-FID; GC-MS	[41]
Leaves	Greece	*α*-pinene (9.4–24.9%), terpinen-4-ol (6.8–10.6%), limonene (9.0–17.8%) and other minor fractions.	GC-MS	[42]
Leaves	Spain	*β*-myrcene (19%), *α*-terpineol + terpinen-4-ol (15%) and other minor fractions.	GC-MS	[43]
Unripe fruit		*β*-myrcene (54%), *α*-pinene (22%).	GC-MS	
Ripe fruit		*β*-myrcene (19%), *α*-pinene (11%), δ-3-carene.	GC-MS	
Leaves	Egypt	δ-3-carene (65%), sesquiterpene alcohols (4%).	GC-MS	[47]
Mastic gum	Greece	*α*-pinene (41–67%), limonene, *β*-linalool an perillene	GC-MS	[48]
Mastic Gum	Turkey	*α*-pinene (56.2–70%) myrcene (2.5–20.1%), and *β*-pinene (2.5–3.1%)	GC-FID, GC-MS, Chiral GC	[49]
Leaves	Greece	Myrcene (20.6%), germacrene D (13.3%), E-caryophyllene (8.3%) and other minor fractions.	GC-MS	[65]
Leaves and twigs	Sardinia (Italy)	Terpinen-4-ol (25.2%), *α*-phellandrene (11.9%), *β*-phellandrene (10.2%) and other minor fractions.	GC-FID, GC-MS	[66]
Female flowers		*α*-limonene (28.7%), germacrene-D (23.7%), elemol (6.7%) and other minor fractions.	GC-MS	
Leaves of male plants at flowering		*α*-limonene (18.8%), germacrene-D (13.1%), *β*-caryophyllene (8.8%) and other minor fractions.	GC-MS	
Leaves of female plants at flowering		Germacrene-D (20.7%), δ-cadinene (15.6%), *β*-caryophyllene (12.1%) and other minor fractions.	GC-MS	
Ripe fruits		*β*-myrcene (75.6%), *α*-pinene (12.6%), *α*-limonene (3.2%) and other minor fractions.	GC-MS	
**Extracts**				
**Plant material**	**Origin**	**Main components**	**Test Assays**	**Ref.**
Leaves/metanol-water	Algeria	Total phenolic acids 429.58 ± 3.26 (mg CatE/g E), total flavonoids 139.38 ± 3.11 (mg RutE/g E), total tannin 142.56 ± 2.60 (mg TAE/g E).	UPLC-DAD	[32]
Aereal parts/metanol water	Greece	Total phenolic acids (mg gallic acid/g plant) 483 ± 2.7 (winter season), 588 ± 32.7 (Spring season), 581(a) ± 14.0 (Summer season)	Folin-Ciocalteu	[42]
Leaves/ethyl acetate and methanol	Italy	3,5-O-digalloyl quinic acid (26.8 ± 0.15 mg/g DW), 3,4,5-O-trigalloyl quinic acid (10.3 ± 2.45 mg/g DW), 5-O-galloyl quinic acid (9.6 ± 2.45 mg/g DW) and other minor fractions.	HPLC-DAD, HPLC-MS, NMR	[59]
Leaves/water ethanol	Croatia	Total Phenolic Content (mg/g) 108.14 ± 2.12	Folin ciocalteu method	[60]
Fruits/water ethanol		Total Phenolic Content (mg/g) 41.14 ± 0.76		
Fruits/methanol-water	Tunisia	Gallic acid, Tyrosol, 4-hydroxyphenylacetic acid, Vanillic acid, p-coumaric acid, Ferulic acid, Trans-4-hydroxy-3-methoxycinnamic acid, o-coumaric acid, Oleuropein aglycon, Luteolin, Kaempferol, Naphtoresorcinol, Salycilic acid, Pinoresinol, Apigenin, Coumarin, Carnosic acid and Trans cinnamic acid.	HPLC-DAD	[54]
Fruits/metanol-water		Total phenolic acids 205.79 ± 6.51 (mg CatE/g E), total flavonoids 6.28 ± 1.04 (mg RutE/g E), total tannin 216.74 ± 2.605.31 (mg TAE/g E).		
Leaves and stems/metanol water	Algeria	Total phenolic acids 114.95 ± 6.25 (mg GAE/1 g extract), total flavonoids 25.212 ± 2.13 (mg QE/g extract)	Folin-Ciocalteu, ACCM, TLC	[67]

Acronyms. Gas chromatography–flame ionization detection (GC-FID); Gas chromatography–mass spectrometry (GC-MS); Ultra-Performance Liquid Chromatography with Diode-Array Detection (UPLC-DAD); High-Performance Liquid Chromatography with Diode-Array Detection (HPLC-DAD); High-Performance Liquid Chromatography-Mass Spectrometry (HPLC-MS); Nuclear Magnetic Resonance (NMR); Aluminum chloride colorimetric method (ACCM); Thin Layer Chromatography (TLC).

## 3. In-Vitro Data on Oral Planktonic Microorganisms

Many laboratory studies attest to the antimicrobial activity of PlL derivates. The agar disc diffusion test (ADD) and the micro-broth dilution method determining the minimal inhibitory concentration (MIC) have been usually applied to quantify this capacity. More rarely, the minimal bactericidal/fungicidal concentration (MBC/MFC) and the time-killing assay were selected to measure the antimicrobial potency.

Commonly studied oral microorganisms vary from Gram-positive bacteria (*Streptococcus* spp., *Enterococcus faecalis*) to Gram-negative bacteria (*Porphyromonas gingivalis*, *Fusobacterium nucleatum*, *Actinomyces* spp. *Tannerella forsythia*), the yeasts *Candida* albicans and *C. glabrata*, and sometimes viruses, mainly represented by the herpes simplex virus. Concerning *Streptococcus* sp., many studies demonstrated the high sensitivity of *S. mutans* to PlL [37,68]. Moreover, periodontal pathobionts were strongly inhibited by the agent [36]. Conversely, some papers evidenced any or low antifungal activity of PlL derivates [37,67,69,70,71,72], even if high capacity against the growth of *Candida* as laboratory as clinical isolate sp. was demonstrated very recently using the leaves EO [36].

Regarding the berry oil, Orrù and coworkers [37] found a selective inhibition of the agent toward different species of *Streptococcus* using the MIC and MBC/MFIC. *S. agalactiae*, *S. intermedius*, *S. mutans*, *S. pyogenes* were highly sensitive, and this correlated to the pathogenic profile of the bacteria.

## 4. In-Vitro Data on Biofilm Experiments

Orrù and coworkers also evaluated the capacity of PlL berry oil toward *Streptococcus* sp. in biofilm [37]. The results showed that the sensitivity of the bacteria was species-specific. Two probiotic strains of *S. salivarius* K12/M18 were non-sensitive to the oil. Conversely, the sensitivity of the pathogens *S. agalactiae*, *S. intermedius*, *S. mutans*, and *S. pyogenes* was correlated to the fatty acid profile of the oil. Chromatographic analysis of the growth medium containing the oil showed a significant increase in oleic, palmitic, and linoleic acids, which are already known for their antimicrobial capacity. In this context, it was said that PlL berry oil selectively inhibits the growth of pathogens by the fatty acid metabolic pathway while preserving the vitality of beneficial bacteria.

## 5. Research Related to Dental Caries

Many studies pointed to the activity of PlL against the cariogenic biofilm by carrying out in vitro and in vivo investigations (Table 2, Table 3 and Table 4). Most studies focused on the capacity to antagonize *S. mutans* as a key bacterium in the multifaceted events leading to the demineralization process [73,74]. Additional research included acidogenic microorganisms, i.e., *Lactobacillus*, acidic strains of *non*-*mutans Streptococci*, and *Actinomyces* [73]. Together these bacteria contribute to an increase in the biomass and extracellular polymeric substance (EPS) constituting the cariogenic biofilm.

In this context, one of the most studied materials has been the mastic resin, commonly used against toothache in popular medicine. The ability of mastic against the oral *streptococci* was mainly evaluated using the ADD, MIC, and MBC (Table 2 and Table 3).

In this field, Aksoy and coworkers [76] assayed the capacity of mastic chewing in inhibiting the growth of *Mutans* streptococci and *Lactobacillus* spp. in saliva [76,84]. After mastic chewing, significantly lower bacteria were found in saliva samples in comparison to the control (*p* < 0.001). The results were related to the chemical content of terpinen-4-ol and *α*-terpineol in the mastic gum, which conferred relevant biological properties to the resin. The result asserted that mastic is a useful agent in preventing the disease.

The effectiveness of chewing mastic gum against the salivary counts of *S. mutans* was also reported by Preethi [85], who tested the capacity of the resin gum in a group of children compared to xylitol chewing gum. Mastic gum showed a statistically significant reduction of the bacterium in saliva in comparison to xylitol gum. The property to lower the cariogenic bacteria count in saliva was also attributed to the presence of PlL extract as a component of herbal toothpaste [87].

Further studies investigated the matter. Karygianni and coworkers [77] proved the antibacterial activity of three terpenoid acids derived from PlL resin extract: 24Z-isomasticadienolic acid, oleanolic acid, and oleanonic aldehyde. The authors demonstrated that the acids had an inhibitory effect against *S. mutans* and other *streptococci* (*S. sobrinus*, *S. oralis*). Among the three compounds, the most active was the pentacyclic triterpenoid oleanolic acid, which not only possessed the greatest effect against *S. mutans* but also inhibited the yeast *C. albicans* as a cause of increased aggressiveness in cariogenic biofilm [74]. The antimicrobial mechanism of oleanolic acid was shown to disrupt the cell envelope leading to cell death. The interplay between natural compounds and *Candida* has been further elucidated using transmission electron microscopy (TEM) [88,89].

This study further supports the anti-cariogenic effect of the fatty acids of PlL oil, as reported by Orrù and coworkers in planktonic and biofilm cultures [37].

Further analysis attested to the antimicrobial activity of PlL EO of leaves targeting *S. mutans* and *Lactobacillus acidophilus* in adjunction to *E. faecalis* and *F. nucleatum* [90].

## 6. Research Related to Periodontal Disease

Many studies studied the capacity of PlL derivates against periodontal pathogens (Table 2, Table 3 and Table 4). Among them, several studies aimed to elucidate the antagonism of mastic toward *P. gingivalis* as a key bacterium in periodontal disease [77,91,92]. The antimicrobial activity of water-ether extracts of mastic was addressed by the presence of three major triterpenoid acids [77]. To validate this, 24Z-isomasticadienolic acid, oleanolic acid, and oleanonic aldehyde were screened as isolated fractions against *P. gingivalis*. *F. nucleatum* and *Parvimonas micra.* The acids showed different inhibitory capacities toward the bacteria, been the oleanolic acid the most effective.

Further, the capacity of mastic chewing against the total anaerobic bacteria in saliva was compared to that of a placebo gum [93]. The study was conducted on 20 young, healthy subjects. Unstimulated saliva was collected as a baseline sample. Then, each group used the assigned chewing gum for 10 min, and saliva was collected again at the end of 1, 2, 3, and 4 h after the chewing. The cultured saliva samples demonstrated a significant reduction in the total amount of anaerobic bacteria in the mastic group compared to the control.

Additionally, the sensitivity of *P. gingivalis* to PlL fruit extract and the leaves extract was carried out in one research [94]. Results demonstrated higher inhibition activity of the fruit extract toward the bacterium than the leaves extract. In the same study, the authors tried to identify the more potent antimicrobial fraction in the fruits fingerprint, which was mainly composed of gallic acid, catechin, 3,4-dihydroxyhydro-cinnamic acid, benzoic acid, salicylic acid, and luteolin. However, synergic activity between all the fractions was proved as responsible for the antimicrobial potency of the fruit extract.

Regarding the EO, the antimicrobial activity of PlL leaves oil was recently reported against *P. gingivalis* and toward *T. forsythia* and *F. nucleatum* [36]. In particular, the MIC of the PlL leaves EO against *P. gingivalis*, *T. forsythia* ranged between 3.13 mg/mL. The result was addressed to distinguish between the pharmacological fractions of the terpenes characterizing the oil chemotype, which included *α*-pinene (16.89%) and terpinen-4-ol (16.49%), in addition to the presence of limonene (3.89%) *β*-myrcene (0.87%), the sesquiterpenes (Z)-caryophyllene (1.39%), and (E)-caryophyllene (0.07%).

Several studies also report on the biocompatibility of PlL extracts and oils as antimicrobial compounds. Cytotoxicity testing was conducted for a variety of cells, i.e., human immortalized keratinocytes [94], human gingival, periodontal ligament fibroblasts, gingival keratinocytes, and dysplastic oral keratinocytes [36], human osteoblastic cell lines, and mouse fibroblast cell lines [78]. However, any potential toxic effect on cell lines was ascribed to PlL derivates in the evaluations in addition to the antimicrobial capacity. Conversely, an increase in the viability of fibroblasts was reported by the WST-1 metabolic assay testing PlL EO [33]. Thus, all the reports show that the high biological character of PlL derivates occurs when used as oral antimicrobials.

## 7. Oral Halitosis

Mastic gum is an ancient remedy against oral halitosis [79], a common clinical sign of periodontal disease, dry mouth, smoking, alcohol consumption, and stress [74,95]. Halitosis derives from VSCs greatly produced by the Gram-negative anaerobic bacteria supported by the Gram-positive bacteria [7]. Furthermore, *C. albicans* can be involved in the production of VSCs [8], which accumulation in oral surfaces may also contribute to the development of cancer [96].

Several researchers focused on the activity of mastic gum in inhibiting the responsible oral bacteria for the production of VSCs (Table 2 and Table 3). In this context, Sterer and coworkers [86] investigated the capacity of mastic gum as a component of palatal mucoadhesive tablets formulated to antagonize oral malodor and VSC levels. Using the ADD, the authors demonstrated the strength of mastic against *P. gingivalis*, *C. albicans*, and *S. mutans*, as pathogens involved in oral halitosis.

## 8. *Candida* Infection

The activity of PlL derivates against the yeasts in general, and *Candida* sp. in particular, has been widely investigated (Table 2 and Table 3).

A high inhibiting effect on *C. albicans* was reported by examining the extracts. In this regard, the antifungal power was associated with the anti-antioxidant potency of an extract, which was further related to an increase in flavanols concentration [66,71,72,80,81,82,83]. Further evidence concerns the amount of tannic acid. This phenol was more effective than nystatin and amphotericin, commonly used as antifungal agents. Moreover, linalool and *α*-terpineol were identified as potent antifungals in mastic water obtained by the steam distillation of mastic resin [97]. Also, the extraction solvent was reported as crucial in increasing the antifungal and antioxidant capacity of the agent [82].

Additional activity against *C. albicans* was reported by a poly-herbal mixture where PlL water extract was included as an ingredient of the paste. In this case, the significant capacity was comparable to that of fluconazole [98].

Conflicting data regarding the antifungal strength of PlL EO [37,71]. However, low MICs against *C. albicans* and *glabrata*, clinical isolate sp., were noticed by assaying PlL leaves EO [36]. The authors addressed such activity to the presence of six pharmacological terpenes, which were recovered in a concentration above 0.05%. In the same research, the authors demonstrated the capacity of the oil to inhibit COX2 and lowering LOX, supposing the capacity of the EO to antagonize the level of PGE2. In fact, the aggressiveness of *Candida* is strongly related to the ability to produce high quantity of fungal PGE2, which also exhibits cross-reactivity with that of the host [99,100,101]. The data could attest to important properties of the leaves EO against the yeast as COX-2 and LOX inhibitors are recommended to antagonize the biofilm development, resistance, and invasion of *Candida*.

## 9. Oral Virus Infection

Only two studies assessed the capacity of PlL derivates against HSVs. With the exception of the ethyl acetate extract, Suzuki and colleagues [102] reported the capacity of different PlL extracts. Moreover, Bouslama [103] described a significant activity of the methanol extract of the PlL stem against HSV-2. An explanation of these data could be found in the richness of polyphenols of PlL extracts and in their capacity to recover hormesis mechanisms.

## 10. Clinical Studies

A few numbers of studies documented the effectiveness of PlL derivates clinically. It was pointed to validate the popular capacity of mastic to antagonize toothache and gingival inflammation. With this intent, in a double-blinded, randomized study placebo-controlled, Takahashi and coworkers studied the ability of mastic chewing gum against tooth plaque formation and gingiva inflammation [93]. The study was conducted on 20 young volunteers with healthy periodontal. The plaque and gingival indices were recorded at baseline and at the end of the trial, during which a piece of mastic gum or of the placebo gum was chewed as the only oral hygiene device. The clinical data showed that the degree of plaque accumulation and gingival inflammation were significantly lower in the group of mastic chewing in comparison to the control.

Mastic resin was also included in a mixture of herbal polymers forming palatal mucoadhesive tablets in antagonism to oral malodor production and VSCs [86]. Results showed a significant reduction in malodor scores and VSC levels in the experimental group in comparison to the control.

## 11. Discussion

This review aimed to summarize the scientific knowledge on the antimicrobial capacity of PlL biomolecules toward oral biofilm-associated diseases. In view of the data, a second aim of this work was to suggest a possible use of PlL derivates as a health cure agent. This is in accordance with the need to find novel agents able to combat biofilms causing resistant bacteria and diseases.

Regarding our search, we found a great number of investigations that, together with the antimicrobial capacity, simultaneously reported on the wide biocompatibility, and the great antioxidant and anti-inflammatory properties possessed by PlL biomolecules. The evidence that natural molecules can exhibit several benefits at the same time strongly demonstrates their various working abilities in comparison to chemically formulated antimicrobial agents. In fact, natural antimicrobials can antagonize pathogens and behave in such a way as to regulate the altered environment. Furthermore, natural compounds can target bacteria by virulence/detrimental factors with biocide-free effects rather than the processes involved when employing chemical agents, leading to sustainable management of biofilm challenges. In such a way, natural antimicrobials can successfully counteract the alarming scenario of resistant bacteria.

The potential potency of polyphenol extracts has led to increasing scientific interest. Several trials confirm the value of these molecules in maintaining and recovering healthy status due to the management of the cell homeostatic systems. Polyphenols act as stressors to the exposed animal cells activating cell defense systems to control the redox environment, the proteostatic and metabolic homeostasis, the organelle turnover, and the inflammatory response [103]. As disturbance in homeostasis is also strongly related to the development of oral diseases, the high content of polyphenols of PlL extracts is particularly appealing in the search for new types of oral health care agents. Regardless of the parts of the plant they derive from, the extracts are among the richer extracts in polyphenols described in the literature. In the leaf extract, the molecules had a range of 7.5% of leaf dry weight, where the amount of galloyl quinic acid derivates, together with flavonol glycosides, greatly represented by myricetin derivates, and quercetin, in addition to anthocyanins are the main expressed agents. In addition to the antimicrobial properties, these chemical fractions confer to the extracts high antioxidant and anti-inflammatory capacity, together with anticancer, hepatoprotective, and hypercholesteremic characteristics.

Reviewing the literature, we found that the majority of the studies focused attention on the inhibitory effect of PlL derivates against cariogenic and periodontal bacteria in addition to *C. albicans*. Discordance data have been collected regarding *S. mutans* and other cariogenic microorganisms. The mastic resin and the EOs were the most active compounds against *S. mutants* and cariogenic bacteria in general. While, with the exception of mastic extract, the polyphenols of the leaves and the fruit were unable to block the bacteria.

Considerable antiseptic activity is reported against *P. gingivalis* and other periodontal pathogens by using all of the various derivates. Several studies attempted to isolate the most active biomolecules against *P. gingivalis*, but a synergic effect between all the fractions was necessary to stop the viability of the bacterium. Regarding the extract, laboratory studies demonstrated the greater efficacy of the fruit extract in comparison to that of the leaves. However, the latter showed the highest antioxidant capacity, which was related to a major content of flavonoids. A high anti-ROS property, and anti-inflammatory ability are still necessary to recover altered oral homeostasis by periodontal dysbiosis, and both these abilities work independently of the MIC. In fact, at dose ranges lower than the ascertained MIC, gallic acid-rich extract was able to downregulate *gingipains* expression by *P. gingivalis*, significantly affecting the release of inflammatory genes (i.e., COX-2) [104]. These data are consistent with the finding that sub-lethal concentrations of polyphenols have a proper time-dependent antibiofilm effect [60,105].

Concerning *Candida* sp., a great number of references cite PlL extracts as anti-*Candida* agents, whereas the EOs and mastic resin showed contrasting data. The high content of flavonoids in the extract is reported to be a determinant in eliminating the yeast. Among these biomolecules, the phenol tannic acid demonstrated higher antifungal activity than chemicals usually applied as chemotherapy agents. A considerable number of studies stressed ascertaining the MIC capacity of the polyphenols against cultivated species of *Candida*. While clinical species were rarely evaluated by studies, all of them analyzed biofilm. This suggests the need to investigate the capacity of biomolecules in such complex communities characterized by intrinsic resistance to host immune response and antimicrobial drugs. In this regard, in addition to cell growth inhibition, polyphenol properties are to be considered. Catechin and proanthocyanidins were found to react with spores and hyphae of the pathogenic fungi, interfering with quorum-sensing behavior. Additionally, natural phenols, as anti-inflammatory substances, are able to reduce the level of COX enzymes, thus interfering with the release of prostaglandin E2 (PGE2) expression, which is necessary to *Candida* in the formation of biofilm [106]. Given this evidence, a double effect can antagonize *Candida* oral infections using PlL polyphenols: a direct activity against the yeast and an indirect efficacy against its virulence, in addition to the absence of cytotoxicity, as it was largely verified in the studies.

Finally, this review has documented several clinical studies on the effectiveness of PlL derivates. However, the need to increase clinical knowledge demands trials to better understand the behavior of PlL polyphenols in oral health and diseases.

## 12. Conclusions

The positive findings regarding the inhibition of periodontal pathogens and *C. albicans*, together with the antioxidant activity and the reduction of the inflammatory responses, would strongly suggest the use of PlL polyphenols in the prevention and/or reversal of intraoral dysbiosis. In fact, the extract appears to be a significantly more effective agent than the other PlL derivates.

Toothpaste, mouthwashes, and local delivery devices could be effective in the clinical management of these oral diseases. Nanotechnology encapsulation will be useful in obtaining stable compounds able to transfer the pharmaceutical capacity of PlL derivates in the prevention and treatment of oral diseases.

## Figures and Tables

**Figure 1 microorganisms-11-01378-f001:**
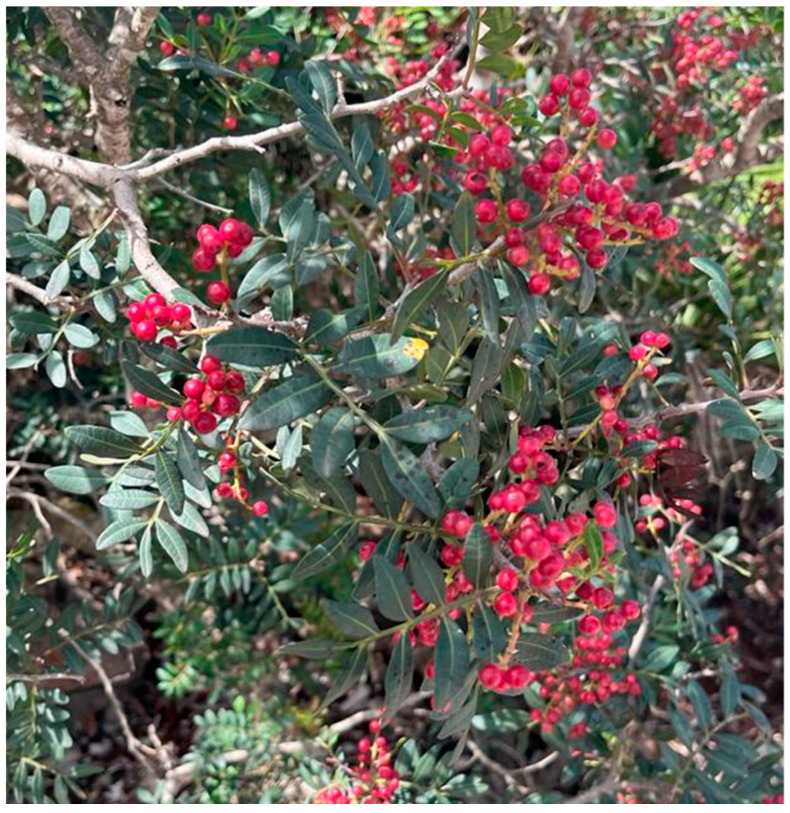
A female plant of the wild-growing shrub *Pistacia lentiscus* L.

**Table 2 microorganisms-11-01378-t002:** Antimicrobial activity of *Pistacia lentiscus* L. oils against the oral pathogens.

Essential Oil						
Origin	Plant Material	Bacteria/Fungi	Origin of Strain	Method	Antimicrobial Activity	Ref.
Greece	Mastic Oil	*Escherichia coli*	Not given	ADD	Active	[27]
Eastern Morocco	Aerial parts	*Streptococcus* spp.	Not given	ADD	Active	[29]
		*E. coli*	Not given		Active	
		*Pseudomonas* spp.	Not given		Active	
Sardinia (Italy)	Leaves	*Streptococcus gordonii*	ATCC 10558	MIC	Active	[36]
		*Actinomyces naeslundii*	ATCC 12104		Active	
		*Fusobacterium nucleatum*	ATCC 25586		Active	
		*Porphyromonas gingivalis*	ATCC 33277		Active	
		*P. gingivalis*	Clinical (*n* = 2)		Active	
		*Tannerella forsythia*	ATCC 43300		Active	
		*T. forsythia*	Clinical (*n* = 2)		Active	
		*Candida albicans*	Laboratory		Active	
		*C. albicans*	Clinical (*n* = 2)		Active	
		*Candida glabrata*	Laboratory		Active	
		*C. glabrata*	Clinical (*n* = 2)		Active	
Sardinia (Italy)	Fruits	*Streptococcus pyogenes*	Clinical	ADD, MIC	Non active	[37]
		*Streptococcus mutans*	Collection		Non active	
		*E. coli*	ATCC 25922		Low activity	
		*C. albicans*	Clinical		Non active	
		*C. glabrata*	Clinical		Non active	
Greece	Resin, leaves, twigs	*E. coli*	ATCC25922	ADD, MIC	Active	[65]
		*C. albicans*	Not given		Active	
Tunisia	Fruits	*E. coli*	Not given	ADD, MIC, MBC	Non active	[71]
		*C. albicans*	Not given		Non active	

**Table 3 microorganisms-11-01378-t003:** Antimicrobial activity of *Pistacia lentiscus* L. extracts against the oral pathogens.

Extracts						
Origin	Plant Material/ Solvent	Bacteria/Fungi	Origin of Strain	Method	Antimicrobial Activity	Ref.
Greece	Mastic gum/ethanol	*E. coli*	Not given	ADD	Non active	[27]
Lybia	Leaves and stems/water extract/ethyl acetate/ethanol	*C. albicans*	Laboratory	ADD	Active	[75]
Sicily (Italy)	Aerial parts/ethanol and water	*E. coli*	ATCC 35218	MIC, MBC, MFC	Active	[69]
		*C. albicans*	Clinical (*n* = 18)		Active	
		*C. glabrata*	Clinical (*n* = 11)		Active	
Tuscany (Italy)	Leaves/ethyl acetate and methanol	*C. albicans*	Clinical	ADD, MIC, MFC	Non active	[70]
		*C. glabrata*	Clinical		Non active	
Tunisia	Fruits/water methanol	*E. coli*	Not given	ADD, MIC, MBC	Active	[71]
		*Pseudomonas* *aeruginosa*	Not given		Active	
		*C. albicans*	Not given		Non active	
Algeria	Leaves and stems/water methanol	*E. coli*	Not given	ADD, MIC	Non active	[67]
Algeria	Leaves/ethanol and ethyl acetate	*Enterococcus faecalis*	ATCC 49452	MIC, MBC, ADD	Low activity	[72]
		*E. coli*	ATCC 25922		Low	
		*P. aeruginosa*	ATCC 27853		Active	
		*C. albicans*	ATCC1024		Active	
Turkey	Mastic/chloroform; acetone/ petroleum ether/ethanol	*S. mutans*	ATCC27351	ADD, MIC	Active	[76]
Greece	Mastic/ethanol; methanol/dichloro-methane/ cyclohexane	*S. mutans*	DSM 20523	MIC, MBC	Active	[77]
		*Streptococcus sobrinus*	DSM 20381		Active	
		*E. faecalis*	ATCC 29212		Active	
		*E. coli*	ATCC25922		Active	
		*P. gingivalis*	W381		Active	
		*Prevotella intermedia*	ATCC 25611		Non active	
		*F. nucleatum*	ATCC 25586		Active	
		*C. albicans*	Laboratory		Active	
Germany	Mastic/ethanol	*P. gingivalis*	DSMZ 20709	ADD	Active	[78]
		*Aggregatibacter actino*-*mycetemcomitans*	DSMZ No. 11123		Active	
		*P.intermedia*	DSMZ No. 20706		Active	
		*F. nucleatum*	DSMZ No. 20482		Active	
		*S. mutans*	DSMZ No. 20523		Active	
Israel	Mastic/methanol	*P. gingivalis*	ATCC 53977	ADD	Active	[79]
Algeria	Leaves/ethanol	*P. aeruginosa*	22212004	ADD	Active	[80]
		*E. coli*	5044172		Active	
		*C. albicans*	444		Active	
Algeria	Leaves/ethanol	*C. albicans*	Clinical	ADD, MIC	Active	[81]
Sudan	Leaves and stem/water, ethanol, petroleum ether, ethyl acetate, n-butanol	*E. coli*	ATCC 25922	ADD	Active	[82]
		*P.aeruginosa*	ATCC 27853		Active	
		*C. albicans*	ATCC 7596		Active	
Morocco	Leaves/dichloromethane-ethylacetate ethanol-methanol-water	*E. coli*	Not given	ADD, MIC	Active	[83]

**Table 4 microorganisms-11-01378-t004:** Clinical trials evaluating *Pistacia lentiscus* L. derivates against oral diseases.

Reference	Vector	Assessment	Sample Characteristics	Experimental Time	Result and Conclusions
[65]	Toothpaste based mastic vs. control	Salivary CFU of *S. mutans* and *Lactobacillus*, plaque index	60 healthy subjects	4 weeks	Any differences in the groups
[72]	Mastic chewing gum vs. control	Salivary total CFU, plaque index, gingival index	20 healthy young subjects	10 min	Greater antibacterial and antiplaque effects
[76]	Mastic gumvs control	Salivary *S. mutans* CFU and mutans Streptococci	25 healthy young subjects	7 days	Greater antibacterial effect to *S. mutans* and Mutans streptococci
[84]	Mastic gum	Salivary CFU of S. mutans, lactobacilli, viable bacteria	25 subjects with orthodontic fixed appliances	15 min	Greater antibacterial effect against *S. mutans* and Mutans streptococci
[85]	Mastic gum vs. control	Salivary CFU of *S. mutans*	60 healthy subjects	15 min	Significant anticariogenic effect
[86]	Palatal tablets vs. control	Odor judge score, sulfide monitor	56 healthy young subjects	60–120 min	The tablets containing mastic gum was effective in malodour and VSCs level

## Data Availability

Data is contained within the article.

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
