# Peer review of "Antimicrobial Efficiency of Pistacia lentiscus L. Derivates against Oral Biofilm-Associated Diseases—A Narrative Review"

_microorganisms, 2023, doi:10.3390/microorganisms11061378_

Round 1

Reviewer 1 Report

The current manuscript entitled "Antimicrobial efficiency of Pistacia lentiscus L. derivates against the oral biofilm-associated diseases – a narrative review", contain good information about the chemical composition and biological profile of Pistacia lentiscus L.

     Therefore, the manuscript is suitable to be published after minor revisions. Here are my suggestions and comments addressed to the authors:

·        The chemical composition of the essential oil of different parts of Lentisk has been well studied by several authors, so the authors should enrich the part of the chemical composition with other more recent works.

·        The same remark for the quantification of total polyphenols, flavonoids and tannins.

Minor editing of English language required

Author Response

Dear reviewer, thank you for your comments and suggestions. We increased the text in the requested parts.

Best regards.

Reviewer 2 Report

Peer-reviewed work of the authors   Milia  et al., presents a broad overview of the antimicrobial activity of plant compounds from Pistacia lentiscus L.  and  it is  a review based on the nowadays  literature. Undoubtedly  the manuscript  will  be of interest to readers and can be published in the “Microorganisms”.

I consider  that English  should be  corrected. Some sentences in the text need to be rewritten, because they do not quite convey the meaning.  

For example  -The compounds derive by the activity of Gram-negative anaerobic bacteria supported by Gram-positive bacteria and fungi [7,8].

Author Response

Dear reviewer, thanks for your comments and suggestions. We provided a full revision of the English form. We also corrected the orthographic mistakes.

Sorry for these misunderstanding.

Thank you very much.

Reviewer 3 Report

The choice of topic is timely, the tables are easy to understand. The authors cite more than 100 articles, including recent literature.

Please check the attached file for detailed comments.

English language typos and grammatical errors can be observed in the manuscript, they need to be corrected.

Author Response

Dear reviewer,

thank you very much for your comments and suggestions. We increased the text in regard of biofilm and corrected the table you have kindly indicated. We also provided a full revision of the English form.

 Best wishes.